# Genomic Prediction in a Self-Fertilized Progenies of *Eucalyptus* spp.

**DOI:** 10.3390/plants14101422

**Published:** 2025-05-09

**Authors:** Guilherme Ferreira Melchert, Filipe Manoel Ferreira, Fabiana Rezende Muniz, Jose Wilacildo de Matos, Thiago Romanos Benatti, Itaraju Junior Baracuhy Brum, Leandro de Siqueira, Evandro Vagner Tambarussi

**Affiliations:** 1Department of Forest Science, Soils and Enviroment, São Paulo State University (UNESP), School of Agricultural Sciences (FCA), Av. Universitária, Botucatu 18610-034, SP, Brazil; guilherme.f.melchert@unesp.br; 2Department of Plant Production, São Paulo State University (UNESP), School of Agricultural Sciences (FCA), Av. Universitária, Botucatu 18610-034, SP, Brazil; ferreira.fmanoel@gmail.com; 3Suzano S.A., Jacareí 12340-010, SP, Brazil; fabiana.muniz@suzano.com.br (F.R.M.); jwmatos@suzano.com.br (J.W.d.M.); tbenatti@suzano.com.br (T.R.B.); itarajubrum@suzano.com.br (I.J.B.B.); lsiqueira@suzano.com.br (L.d.S.)

**Keywords:** genomic selection, SNP, quantitative genetics, tree breeding

## Abstract

Genomic selection in *Eucalyptus* enables the identification of superior genotypes, thereby reducing breeding cycles and increasing selection intensity. However, its efficiency may be compromised due to the complex structures of breeding populations, which arise from the use of multiple parents from different species. In this context, partial inbred lines have emerged as a viable alternative to enhance efficiency and generate productive clones. This study aimed to apply genomic selection to a self-fertilized population of different *Eucalyptus* spp. Our objective was to predict the genomic breeding values (GEBVs) of individuals lacking phenotypic information, with a particular focus on inbred line development. The studied population comprised 662 individuals, of which 600 were phenotyped for diameter at breast height (DBH) at 36 months in a field experiment. The remaining 62 individuals were located in a hybridization orchard and lacked phenotypic data. All individuals, including progeny and parents, were genotyped using 10,132 SNP markers. Genomic prediction was conducted using four frequentist models—GBLUP, GBLUP dominant additive, HBLUP, and ABLUP—and five Bayesian models—BRR, BayesA, BayesB, BayesC, and Bayes LASSO—using k-fold cross-validation. Among the GS models, GBLUP exhibited the best overall performance, with a predictive ability of 0.48 and an R^2^ of 0.21. For mean squared error, the Bayes LASSO presented the lowest error (3.72), and for the other models, the MSE ranged from 3.72 to 15.50. However, GBLUP stood out as it presented better precision in predicting individual performance and balanced performance in the studied parameter. These results highlight the potential of genomic selection for use in the genetic improvement of *Eucalyptus* through inbred lines. In addition, our model facilitates the identification of promising individuals and the acceleration of breeding cycles, one of the major challenges in *Eucalyptus* breeding programs. Consequently, it can reduce breeding program production costs, as it eliminates the need to implement experiments in large planted areas while also enhancing the reliability in selection of genotypes.

## 1. Introduction

The use of molecular markers for individual selection was first proposed by [1] through marker-assisted selection (MAS). Subsequently, the introduction of haplotypic blocks enabled a more accurate association of traits of interest with quantitative trait loci (QTL) [2]. Despite this advancement, the current methodology still presents considerable challenges when applied to the *Eucalyptus* genus. This is primarily attributed to the fact that most productivity-related traits are quantitative in nature and governed by multiple genes with small effects, thereby reducing the efficiency of MAS models [3]. To overcome these limitations, Meuwissen et al. [4] introduced genomic selection, a methodology that enables a more accurate assessment of the effects of quantitative traits. This approach has become an essential tool in breeding programs, enhancing the precision of genetic value estimation and, consequently, improving the decision-making processes of various breeders [5,6,7]. As a result, genomic selection has proven to be more efficient and reduces the number of cycles required for genetic improvement [8].

Genomic prediction models integrate phenotypic and molecular marker information from individuals and/or populations to train models capable of predicting individual performance before field testing [9]. These models maximize the efficiency of genetic gains in breeding programs by enabling a more accurate estimation of genetic variance components, shortening breeding cycles, and allowing for higher selection intensities [10].

However, the efficiency of genomic prediction depends on several factors, including sample size, as a large number of individuals is required for models to effectively capture population variability patterns [11]. Additionally, crucial elements of a well-designed study include the quality and density of genotyping, the degree of relatedness among individuals, and its methodology [12]. These factors directly influence the models’ ability to predict phenotypes with greater accuracy [13,14].

In *Eucalyptus* breeding programs, genomic selection is primarily used to predict the performance of individuals to be cloned, aiming to shorten breeding cycles [15]. While the development of a commercial clone traditionally requires approximately 13 to 14 years, the application of genomic selection has the potential to reduce this period to around 7 years [16].

However, the efficiency of genomic prediction models in *Eucalyptus* is hindered by the use of unstructured populations and multiple species, leading to a complex population structure. This complexity reduces the models’ ability to accurately estimate individual performance, particularly when compared to other crops such as maize, soybean, and rice [17,18].

Given the increasing demand in the forestry market for productive and well-adapted hybrids, the inbred lineage method has emerged as a promising alternative to meet these requirements. This approach enables the exploitation of additive effects through the self-fertilization of individuals, followed by targeted crosses, resulting in highly productive, simple hybrids free from an undesirable genetic load [19]. This method has been widely adopted in crops such as maize, where, since 1910, simple hybrids have been developed by crossing contrasting lines to meet market demands [20].

In the context of genomic selection, the use of inbred lines enhances the efficiency of this technique in the *Eucalyptus* genus because self-fertilization increases homozygosity, restricts genetic variability, and reduces the effective population size [21]. These conditions improve the ability of genomic selection models to capture genetic variability patterns, leading to more accurate and reliable genotype predictions [22].

Therefore, the objectives of this study were (i) to apply genomic prediction in a self-fertilized population composed of different *Eucalyptus* species to identify promising genotypes for future self-fertilization and (ii) to predict genomic estimated breeding values (GEBVs) for diameter at breast height (DBH) in individuals with no known phenotype cultivated in indoor conditions, aiming at the faster development of inbred lines.

## 2. Results

### 2.1. Kinship Analysis

The paternity analysis included a total of 523 individuals. The findings indicate a significant prevalence of self-fertilization, with 326 individuals (62.33%) being selfed, while 197 individuals (37.67%) were attributed to cross-fertilization events. Furthermore, the proportion of selfed versus crossed individuals exhibited variation among families. This inter-family variability hints at factors such as potential pollen contamination in some instances or inherent differences in the propensity for self-fertilization (or resistance to it) across parental lines (Figure 1).

### 2.2. Decay of Linkage Disequilibrium

The estimation of linkage disequilibrium (LD) was performed exclusively with individuals confirmed to be the result of self-fertilization (Figure 2). The analysis revealed that LD values decreased to below 0.35 at a distance of 300 kilobases (Kb), which indicates an increase in LD as inbreeding progresses.

### 2.3. Genomic Selection

In the genomic selection analysis, the GBLUP (Genomic Best Linear Unbiased Prediction) model exhibited the best performance in terms of predictive ability, with a value of 0.488. This was followed by the additive-dominant GBLUP (0.465), HBLUP (0.444), and ABLUP (0.375). These results indicate that the inclusion of genomic information significantly improves predictive ability compared to traditional pedigree-based models.

Among the Bayesian models, predictive abilities ranged from 0.114 to 0.150 (Figure 3), with the Bayes Ridge Regression (BRR) model achieving the highest predictive ability within this group. However, the performance of the Bayesian models was notably lower than that of the frequentist models, particularly GBLUP.

In terms of the Mean Square Error (MSE), the GBLUP model outperformed the other frequentist models, with the lowest MSE value being 13.336. This was followed by the additive-dominant GBLUP (13.992), HBLUP (14.390), and ABLUP (15.498) (Table 1). These results suggest that GBLUP has a lower error associated with its predictions, making it the more accurate model among the frequentist approaches evaluated.

Among the Bayesian models, the MSE values were very similar, ranging from 3.719 to 3.735. The Bayes LASSO model performed slightly better than the other Bayesian models in this respect, as shown in Table 1.

The Bayesian models exhibited lower MSE values. This advantage stems from their ability to account for uncertainty by incorporating prior beliefs and providing posterior distributions for parameter estimates, but this did not translate into higher coefficients of determination (R^2^) and, because the large amount of noise in these models associated with a smaller training set, led to a smaller predictive ability. The highest R^2^ among the Bayesian models was observed in the BRR model (0.055). In contrast, the frequentist models performed significantly better in this regard, with the highest R^2^ found in GBLUP (0.225), followed by HBLUP and additive-dominant GBLUP (0.185), both of which presented very similar values. GBLUP also stood out by having the lowest standard deviation among the models evaluated.

On the other hand, ABLUP showed the lowest R^2^ value of 0.124, further supporting the superior performance of GBLUP among the frequentist models.

Based on the parameters evaluated for all tested models, GBLUP demonstrated the best performance, exhibiting the highest predictive capacity and the lowest MSE. These results suggest that GBLUP is the most reliable model for genomic prediction analyses, making it the optimal choice for this study. Consequently, GBLUP was selected to predict the performances of individuals from the hybridization orchard, with the goal of ranking them for selection. This approach ensures that individuals with the most promising genetic potential are prioritized for further breeding and selection efforts.

## 3. Discussion

### 3.1. Linkage Disequilibrium

The increase in linkage disequilibrium (LD) observed in the study’s population can be attributed to two primary factors. Firstly, the parents used to generate the seeds are commercial clones, meaning they are derived from already improved populations. These populations typically exhibit reduced genetic variability and smaller effective population size, which can lead to higher LD among the selected parents [23]. Secondly, the process of self-fertilization further contributes to the rise in LD. During self-fertilization, heterozygous loci segregate into homozygous genotypic classes, which increases the proportion of fixed alleles in the genome [24]. These fixed alleles result in high LD, as the same genotypic classes are consistently maintained across generations of self-fertilization to obtain inbred lines. During gamete formation, alleles tend to recombine with each other and become highly linked due to factors such as recombination and selection pressures, leading to linkage disequilibrium [25].

LD plays a significant role in genetic improvement programs, particularly in the production of inbred lines, due to its relationship with homozygosity and genomic stability. As genetic variability within families decreases and variability between families increases, this genomic structure can represent a challenge for clone selection programs that depend on intrafamilial genetic diversity to identify superior genotypes. However, in the context of inbred lines, this characteristic becomes beneficial [26]. One of the main objectives of inbreeding programs is the production of homozygotes, and the increase in LD is directly related to genomic stabilization. Therefore, LD aids in preserving specific allele combinations in highly correlated genomic blocks [23]. This stabilization process promotes the efficient fixation of genes of interest, reduces the need for recombination, and accelerates selection.

Furthermore, an enhanced LD in populations facilitates the more efficient use of genomic models in breeding programs that employ genome-wide selection [7,27]. In highly inbred populations, an increased LD strengthens the associations between markers and quantitative traits, thereby improving the accuracy of genomic predictions [28]. This increased accuracy allows for the efficient identification and fixation of favorable alleles, as well as the strategic planning of crossing schemes between lines, thereby enhancing genetic gains [29]. This greater predictability is particularly beneficial for programs that aim to produce uniform lines for commercial purposes, such as the production of improved clones [8].

### 3.2. Genomic Selection and Prediction

GBLUP demonstrated a superior performance compared to the other models tested, largely due to its use of the genomic kinship matrix. This matrix enabled a more accurate estimation of the genetic relationships between individuals based on molecular markers [27,30,31]. GBLUP is particularly effective in capturing additive genetic variability, making it especially advantageous for polygenic traits. Here, multiple loci of small effect contribute to phenotypic expression [7,31].

In contrast, the lower performance of the Bayesian models tested may be attributed to the way these models handle the distribution of marker effects. Bayesian models assume that many loci have null effects and prioritize those with more significant effects [32]. This assumption reduces their predictive efficiency for polygenic traits, where the effects are distributed throughout the genome. This explains the lower predictive capacities observed in these models, which ranged from 0.114 to 0.150, while GBLUP demonstrated a predictive capacity of 0.488.

A factor that influences these results is that the heritability of the analyzed trait can range from 0.3 to 0.7, depending on the population and experimental design, so the DBH is highly influenced by the environment [10,33]. It is difficult for genomic selection models to capture and understand the genetic variance patterns in a population, even in populations with a lower number of individuals, as in the present study. Thus, genomic selection models can present low predictive accuracies [8]. Another factor influencing these results is the relationship between the models and the size of the database. GBLUP showed greater stability and was less susceptible to overfitting, making it more suitable for smaller databases, which is the case in this study. On the other hand, Bayesian models displayed a greater tendency to overfit, possibly due to their attempt to capture secondary patterns [34,35]. Thus, GBLUP can be considered the most efficient model among those tested.

Previous studies have explored predictive capacities for a range of different species, including *Eucalyptus*. For maize, Windhausen et al. (2012) [36] and de Peixoto et al. (2024) [37] reported predictive capacities ranging from 0.53 to 0.69 and from 0.46 to 0.61, respectively. Similarly, Resende et al. (2012) [7] found predictive capacities between 0.55 and 0.56 when evaluating *Eucalyptus* breeding populations. Conversely, Duarte et al. (2024) [38] reported lower predictive values compared to the present study, ranging from 0.24 to 0.39, and Mphahlele et al. (2020) [39] and Grattapaglia et al. (2011) [40] demonstrated predictive capacities ranging from 0.47 to 0.67 and 0.54 to 0.69, respectively. A common feature across these studies is the use of large samples of over 1000 individuals to improve the training and validation sets and a considerable number of significant markers. This previous work indicates that the use of a large number of individuals is needed to achieve higher predictive accuracy values, which was not the case in the present study.

The lower values in this study can be attributed to the smaller number of individuals sampled compared to the larger populations typically analyzed in the literature. For higher predictive capacity values, a population size of at least 1000 individuals is recommended, combined with a high density of markers of interest [40].

Despite the limited sample size, the predictive capacity values in this study were considered satisfactory. This is likely due to the use of the winsorization technique, which limits the impact of extreme values and reduces distortions in genetic effect estimates [41]. The application of this technique notably improved the predictive capacity of the tested models, particularly the frequentist models, by minimizing the influence of outliers.

The population structure and the effects of inbreeding resulting from self-fertilization also played a key role in the efficiency of genomic selection models. The reduction in the effective population size due to self-fertilization favored the application of genomic selection, as lower genetic diversity can enhance the accuracy of predictive estimates [42]. Self-fertilized populations share a greater proportion of alleles, which reduces genetic variability and facilitates the identification of consistent genetic patterns [43]. Thus, genomic selection models can capture additive effects more efficiently [4,44].

Even in populations with a reduced sample size, greater genetic homogeneity enhances the predictive capacity of the models, maximizing the accuracy of genomic estimates [28]. This factor makes genomic selection particularly valuable in forest improvement programs, where the ability to predict genetic effects accurately is crucial for identifying and fixing superior genotypes.

Heidaritabar et al. (2016) [45] and Liang et al. (2018) [46] reported that inbreeding effects can significantly influence the predictive capabilities of genomic models, a trend that was also observed in the present study. Increased homozygosity, which results from inbreeding, leads to the presence of fixed alleles in the genome [47]. This reduction in genetic variability forms large homozygous blocks with more significant effects, facilitating the identification of genomic regions associated with traits of interest through genomic prediction models [48].

For example, in GBLUP, the covariance matrix between individuals efficiently captures the additive effects through kinship [49]. As homozygosity increases in individuals’ genomes, the number of alleles shared between them also increases, improving genomic predictions [50]. Although Bayesian models tend to show lower predictive accuracy, this effect is primarily due to population size. These models can capture more noise in the variability, but a smaller number of individuals prevents the model from detecting these patterns, thereby reducing its predictive accuracy [51], even with lower MSE values.

Therefore, accurate genomic prediction models, even in populations with reduced sample sizes, can provide significant benefits. These models enable a more accurate estimation of genetic values, offering a more reliable understanding of genetic associations with phenotypic traits of interest. As a result, they allow for greater selection intensity and shorter selection cycles [7,52,53]. This is particularly important in breeding programs for inbred lines, such as those in *Eucalyptus*, which typically require long cycles to produce lines with high inbreeding rates for hybrid production.

Accurate estimated breeding values, particularly those derived from GBLUP models, facilitate the acceleration of the inbred line development process, especially within the genus *Eucalyptus*. This acceleration of breeding cycles is achieved because the availability of predicted individual performance eliminates the need for extensive field trials and subsequent retrieval for cloning [15]. Consequently, breeders can proceed directly to hybridization orchards, thereby expediting the advancement of inbred line development [37].

The reliability of selecting superior genotypes for breeding programs hinges on efficient genomic selection models with high predictive accuracy, such as GBLUP. By providing more accurate genomic estimated breeding values (GEBVs), these models allow breeders to make more informed decisions, leading to more accurate and potentially larger genetic gains per selection cycle [54], which is critical for the improvement of economically important species, such as *Eucalyptus*.

## 4. Conclusions

Frequentist models demonstrated superior efficiency in genotype prediction, exhibiting strong predictive capabilities even with a population size smaller than typically recommended for cross-validation. Among these, the GBLUP model stood out as the most effective. In contrast, among the Bayesian models, BRR proved to be the most effective; however, its performance was significantly lower than that of GBLUP.

The predicted DBH values enabled the identification of the best-predicted individuals. These results are critical for selecting individuals with superior genomic estimated breeding values (GEBVs) for future selfing generations.

The relevance of these results lies in the subsequent steps of breeding programs aimed at developing inbred lines. The early and accurate identification of superior individuals enables us to optimize selection for subsequent generations. Individuals with the highest genetic potential, as shown by their ranking, can be prioritized in both controlled crosses and the production of new self-fertilized generations. This strategy can translate into greater efficiency in the breeding cycle and more consistent and rapid genetic advancement.

## 5. Material and Methods

### 5.1. Study Population

#### 5.1.1. Obtaining Seeds and Paternity Testing

To define the study population, 28 commercial clones were selected, including 26 *Eucalyptus urophylla* × *Eucalyptus grandis* hybrids, one *E. grandis* clone, and one hybrid derived from a cross between *E. urograndis* and a species that remains unidentified. All genotypes had the same number of pollinated flowers and therefore produced the same number of fruits. These clones underwent self-fertilization, yielding between 1 and 33 seeds, with variation coming from the parental genotypes (Table 2). In part, the observed high variation can be attributed to potential seed abortion in certain genotypes. Additionally, the requirement of sufficient DNA extraction for genotyping meant that only seeds capable of providing adequate material were included. This effectively restricted the genomic selection analysis to parents with a higher seed set.

To verify whether the obtained seeds resulted from self-fertilization or crossbreeding, a paternity analysis was conducted using Cervus software 3.0.7 [55]. This analysis aimed to correct the pedigree and determine the proportion of self-fertilized and cross-fertilized individuals within each studied family.

Paternity analysis was based on the ∆ statistic [55], defined as the difference in the Lod score between the top two candidate fathers for each tested genotype. Simulations with 10,000 replicates were performed to establish the significance threshold for ∆, with a confidence level of 80%. A total of 723 SNPs with the highest polymorphism were used in the analysis (Figure 4), with an accepted genotyping error rate of 5%. The high degree of polymorphism is crucial because the substantial allelic diversity exhibited by these markers increases the probability of identifying unique genetic profiles. As a consequence, the power of exclusion and the overall accuracy in establishing biological relationships between individuals is enhanced.

#### 5.1.2. Individuals in the Field and in the Hybridization Orchard

Using the obtained seeds, the population was divided into two groups: field-grown individuals (460) and individuals in the indoor hybridization orchard (62). Due to the limited number of seeds obtained from some self-fertilized parents, only 20 out of the 28 parents were used, with a higher number of seeds per parent selected, for a total of 522 individuals. These field-grown individuals were transplanted in a randomized complete block design with thirty blocks and one plant per plot. Also, eight parents and five commercial clones (control) were randomized at the trial. The trial was established at Jacareí, São Paulo, Brazil. At three years of age, the trees were assessed for their diameter at breast height (DBH), determined by measuring the circumference of each tree at 1.30 m above the ground using a measuring tape. The average DBH was calculated to be 12.25 cm (±4.18 cm).

In the indoor hybridization orchard, 62 self-fertilized individuals that were full siblings of the field-grown individuals were planted in pots to induce early flowering. Later, these indoor individuals were genotyped.

Thus, the individuals in the field trial were used to train and validate the genomic selection models. Once the best-performing model was identified, it was applied to the individuals in the indoor hybridization orchard to predict the most promising genotypes.

### 5.2. Genotyping and Quality Control of SNPs

The field and orchard individuals, along with their respective parents, were genotyped using the Eucalyptus 60K chip [56], which covers more than 60,000 SNPs. For linkage disequilibrium analysis and the construction of genomic matrices used in genomic selection models, quality control criteria were applied to exclude specific markers. Markers with a Minor Allele Frequency (MAF) lower than 0.05 and those with a Call Rate below 0.95 were removed. Consequently, markers with an allele frequency lower than 5% within the population and individuals with more than 95% missing data were excluded from the analyses to avoid bias in the estimates.

### 5.3. Linkage Disequilibrium (LD)

Linkage disequilibrium was estimated based on the difference (D) between the observed frequency of two gametes and the expected frequency. Here, larger differences indicated a higher degree of linkage between the two gametes [57]. Linkage disequilibrium values (LD) between pairs of SNPs in the studied population were calculated using the TASSEL 5 software [58]. The linkage disequilibrium decay graph, as a function of the distance in base pairs (bp), was constructed from the r^2^ values using R, version 4.3.2 [59].

### 5.4. Genomic Selection

Genomic selection was conducted in two stages. For the first stage, the training and validation of the tested genomic selection models (Figure 5) was based on the set of field-grown individuals for which both phenotypic and genotypic information were available, including both selfed individuals and some parents that were used as controls in the field population. Initially, a preliminary analysis of the phenotypic data was performed to investigate and characterize these atypical values, including the visual inspection of box plots and histograms. These outliers could compromise the quality of predictions by capturing a disproportionate share of the model’s variability, thereby masking the true variance present in the population and reducing the accuracy of the estimates. To mitigate this effect, the winsorization technique was applied using a 5% percentile to reduce the impact of outliers. Values below the 5th percentile were replaced with the value of the 5th percentile, limiting the influence of extreme data on the statistical analysis and ensuring greater normality in the data distribution [60].

We tested both the frequentist and Bayesian genomic prediction models. The first frequentist model used was ABLUP [61], where the kinship matrix used in predictions was based on the pedigree (A). The second model was GBLUP [62], which utilized a genomic kinship matrix (G) constructed from the additive effects of markers. The next model tested was the additive-dominant GBLUP [63], which accounted for both the additive and dominant effects of markers in the kinship matrix. Finally, the fourth model was HBLUP [64], which combined the pedigree matrix (A) and the genomic matrix (G) to construct a hybrid matrix (H).

The ABLUP model’s matrix [61] can be defined by the following equation:(1)$$Y=X\beta +Z_{a}a+\varepsilon$$
where Y is the *nx*1 vector of phenotypic values, β
**(px1)** is the vector of fixed effects (blocks and general mean), and a
**(qx1) is** the random vector of additive genetic effects, where a~$N\left(0,A\sigma_{a}^{2}\right)\;$(where A is the pedigree-based additive genetic relationship matrix **(qxq)**, $\sigma_{a}^{2}$ is the additive genetic variance), ε
**is** the vector of residual effects, and ε~ $N(0,I\sigma_{e}^{2})$ (where I is the identity matrix and $\sigma_{e}^{2}$ is residual variance) and X (nxp), Z
**(nxq)** are the incidence matrices for the fixed and random effects.

The GBLUP model’s matrix [62] can be defined by the following equation:(2)$$Y=X\beta +Z_{g}g+\varepsilon$$
where g
**(qx1)** is the random vector of the additive genomic genetic effects, g~$N(0,G\sigma_{g}^{2})$ (where G is the genomic relationship matrix (qxq), being $\sigma_{g}^{2}$ the genomic variance) and $\;Z_{g}$
**(nxq)** is the incidence matrix for the genomic effects.

The GBLUP AD model’s matrix [63] can be defined by the following equation:(3)$$Y=X\beta +Z_{g}g+Z_{d}d+\varepsilon$$
where d
**(qx1)** is the random vector of dominance effects, d~$N(0,G_{d}\sigma_{a}^{2})$ (where $G_{d}$ is the genomic dominance matrix (qxq) and $\sigma_{d}^{2}\;$ is the variance of dominance) and $Z_{d}$
**(nxq)** is the incidence matrix for the dominance effects.

The HBLUP model [64] can be defined by the following matrix equation:(4)$$Y=X\beta +Z_{h}h+\varepsilon$$
where h
**(qx1)** is the vector of additive genetic random effects, h~$N(0,H\sigma_{h}^{2})$ (where H is the hybrid genomic matrix (qxq) and $\sigma_{h}^{2}$ is additive genetic variance) and $Z_{h}$ is the incidence matrix for the random effect (nxq).

HBLUP combines information from the pedigree (A) and the genomic matrix (G), forming the hybrid matrix (H), which enhances the accuracy of genomic prediction by integrating genotypic and pedigree data.

The Bayesian models tested (Bayes Ridge Regression—BRR, BayesA, BayesB, BayesC, and Bayes LASSO) followed this matrix structure:(5)Y=Xβ+ε
where Y is the vector of observed phenotypic values, X is the genotype matrix (nxp), and n is the number of individuals and p the number of markers, and β is the vector of random effects of the markers and ε is the residual error.

The difference between the Bayesian models tested lies in the prior assumptions made for each one. These established parameters assess which marker effects were present in the population and which prior distribution best suited the variability of the data.

The Bayes Ridge Regression (BRR) model [65] assumes that all markers have equal variances, i.e., the effects of the markers are considered homogeneous. This model can be represented by the following formula:(6)$$\beta_{j}\sim N(0,\sigma_{\beta }^{2})$$(7)$$\sigma_{\beta }^{2}\sim Inv-\chi^{2}(\upsilon_{\beta },S_{\beta })$$(8)$$\epsilon \sim N\left(0,\sigma_{e}^{2}\right),\;\;\;\;\;\;\;\sigma_{e}^{2}\sim Inv-\chi^{2}(\upsilon_{e},S_{e})$$
where $\beta_{j}$ is the effects of the markers following a normal distribution with a zero mean and $\sigma_{\beta }^{2}$ variance, $\sigma_{\beta }^{2}$ is the variance of the marker effects following an inverse chi-square distribution ($Inv-\chi^{2}$), $\upsilon_{\beta }$ is the degrees of freedom, $S_{\beta }$ is the scale, and ϵ is the experimental error that follows a normal distribution and variance equal to zero ($N\left(0,\sigma_{e}^{2}\right)$).

The BayesA model [4] assumes that each marker has its own variance, resulting in heterogeneous effects among markers. Its formula is as follows:(9)$$\beta_{j}\sim N(0,\sigma_{\beta j}^{2})$$(10)$$\sigma_{\beta j}^{2}\sim Inv-\chi^{2}(\upsilon_{\beta },S_{\beta })$$
where $\beta_{j}$ is the random effect of each marker (j) and $\sigma_{\beta j}^{2}$ is the specific variance of each marker, which follows an inverse chi-square distribution.

The BayesB model [4] introduces the parameter π, which determines the probability of a marker having a null effect. For markers with non-zero effects, heterogeneity in variances is assumed. This model can be represented by(11)$$\beta_{j\;}\left\{\begin{array}{c} N\left(0,\sigma_{\beta j}^{2}\right),\;with\;the\;probability\;(1-\pi )\\ 0\;\;\;,\;with\;the\;probability\;\pi \end{array}\right.$$(12)$$\sigma_{\beta j}^{2}\sim Inv-\chi^{2}(\upsilon_{\beta },S_{\beta })$$
where $\beta_{j\;}$ is the effects of the markers with the correction π for the probability of a marker having zero effect and $\sigma_{\beta j}^{2}$ is the variance of the non-zero markers, which follow an inverse chi-square distribution.

The BayesC model [66] also uses the π correction to represent the probability of a marker having a null effect. However, its main difference from BayesB is that BayesC assumes all non-null markers have the same variance, i.e., homogeneous effects. Its formulation is as follows:(13)$$\beta_{j}\left\{\begin{array}{c} N\left(0,\sigma_{\beta j}^{2}\right),\;with\;the\;probability\;(1-\pi )\\ 0\;\;\;,\;with\;the\;probability\;\pi \end{array}\right.$$(14)$$\sigma_{\beta j}^{2}\sim Inv-\chi^{2}(\upsilon_{\beta },S_{\beta })$$
where $\beta_{j\;}$ is the effects of the markers with correction π with the probability of a marker having an effect equal to zero and non-zero markers having the same effect and $\sigma_{\beta j}^{2}$ is the variance of the non-zero markers, following an inverse chi-square distribution.

The Bayes LASSO model [67] assumes, a priori, a Laplace distribution (also known as a double exponential distribution). This assumption introduces a penalty on the effects of the markers, favoring sparse solutions, i.e., estimates in which many marker effects are reduced to values close to zero. Its formula is(15)$$\beta_{j}\sim Laplace\left(0,\lambda \right)$$(16)$$\lambda \sim \;Gamma\left(a,b\right)$$
where $\beta_{j}$ is the marker effects following a Laplace distribution and λ is the scale parameter of the Laplace distribution, which controls the intensity of the penalization. It follows a Gamma distribution to incorporate uncertainty.

All nine models, considering their respective prior distributions (Table 3), were tested. To evaluate the efficiency of the models, the k-fold cross-validation method was used [68]. In this method, the database was divided into 10 parts (folds), and in each iteration, one of these parts was removed. Missing values were predicted by the models and later compared with the real values.

The parameters used to evaluate the efficiency of the genomic prediction models were clearly defined as essential for assessing the performance of the models. The following evaluation parameters were used:

1.Predictive capacity (PC): the strength of the relationship between the predicted values and the actual observed values. It quantifies how well the model predicts the genotypes. A PC value closer to 1 indicates a better predictive accuracy of the model. The PC [69] formula is(17)$$PC=\frac{Cov(\hat{y},y)}{\sigma_{\hat{y}}*\;\sigma_{y}}$$
where $Cov(\hat{y},y)$ is the covariance between the predicted values ($\hat{y}$) and the actual values (y), $\sigma_{\hat{g}}$ is the standard deviation of the variance of the predicted values, and $\sigma_{g}$ is the standard deviation of the variance of the actual values.2.Mean square error (MSE): the average squared difference between the predicted values and the actual values. It provides an understanding of how much error exists in the model’s predictions. The closer the MSE is to zero, the better the model is at predicting the values correctly. The MSE [70] formula is(18)$$MSE=\frac{1}{n}\sum_{i=1}^{n}{\left(\hat{y}-y\right)}^{2}$$
where $\hat{y}$ are the predicted values, y are the observed values, and n is the number of observations.3.Coefficient of determination ($R^{2}$): the proportion of variance in the observed data that is explained by the model. It indicates how well the model fits the data. An R^2^ value closer to 1 indicates that the model explains most of the variance in the data and thus is performing well. The formula for R^2^ [71] is(19)$$R^{2}=1-\frac{{SS}_{res}}{{SS}_{total}}$$
where ${SS}_{res}$ is the sum of squares of the residuals (the unexplained variance) and ${SS}_{total}$ is the total sum of squares (the total variance in the data).

## Figures and Tables

**Figure 1 plants-14-01422-f001:**
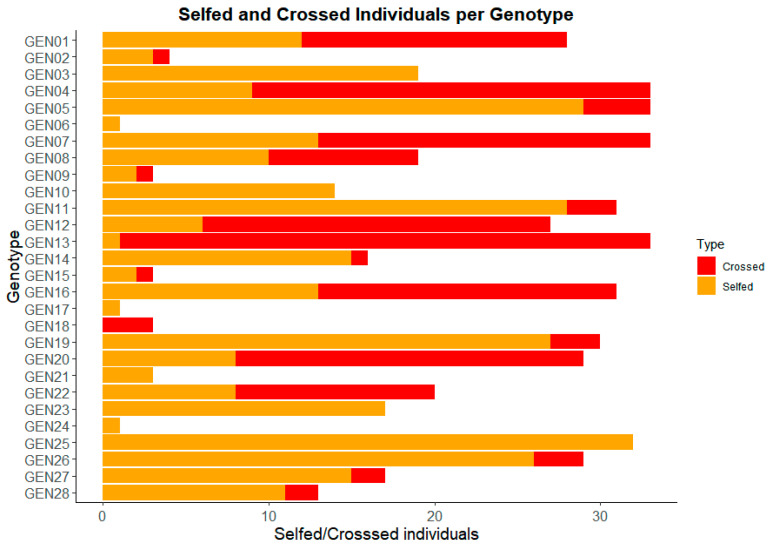
The proportion of selfed and crossed seeds in each selfed genotype of *Eucalyptus* spp.

**Figure 2 plants-14-01422-f002:**
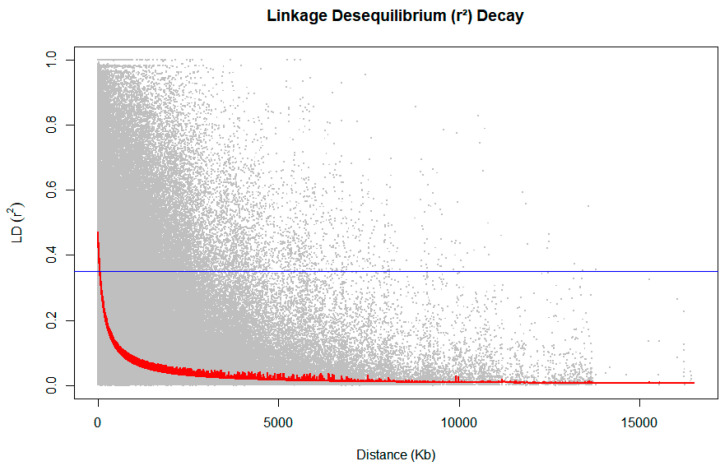
Linkage disequilibrium (LD) decay plot for self-fertilized individuals of *Eucalyptus* spp.

**Figure 3 plants-14-01422-f003:**
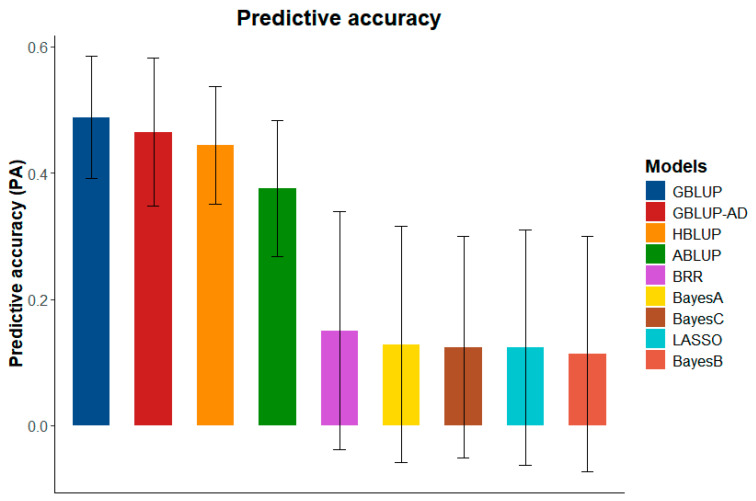
Predictive capacities and standard deviations for genomic prediction models, evaluated by k-fold cross-validation, using data from self-fertilized *Eucalyptus* spp. population at 36 months old.

**Figure 4 plants-14-01422-f004:**
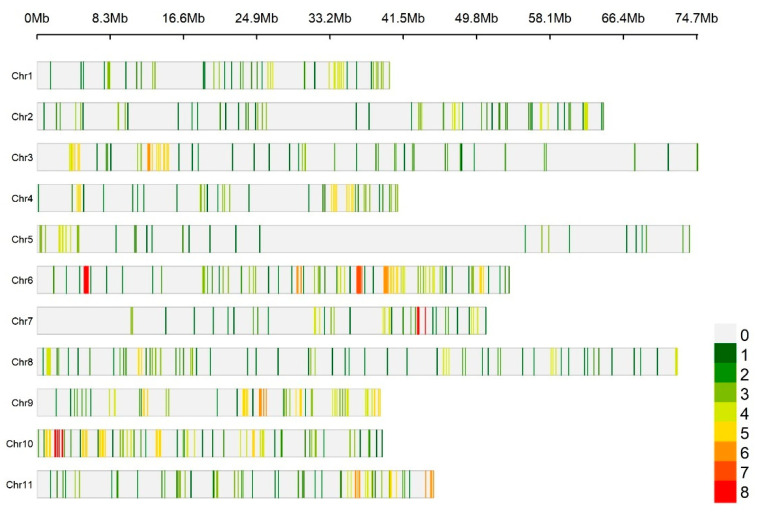
Position in the genome of the 723 polymorphic markers used in the parental analysis in Cervus 3.0.7. in a selfed population of *Eucalyptus* spp.

**Figure 5 plants-14-01422-f005:**
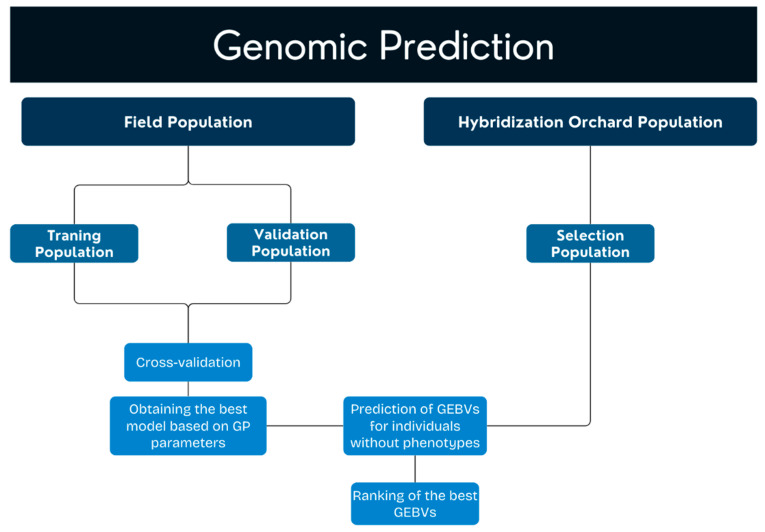
Genomic prediction scheme for field and hybridization orchard populations of *Eucalyptus* spp.

**Table 1 plants-14-01422-t001:** Mean Squared Error (MSE) and coefficient of determination (R^2^) of four frequentist models and five Bayesian models for the trait DBH at 36 months of age in a self-fertilized *Eucalyptus* spp. population.

	Parameters	
Models	EQM	R^2^
GBLUP	13.336 (±1.930)	0.225 (±0.105)
GBLUP-AD	13.992 (±2.565)	0.187 (±0.145)
HBLUP	14.390 (±1.408)	0.185 (±0.077)
ABLUP	15.498 (±1.986)	0.124 (±0.095)
BayesA	3.721 (±0.222)	0.048 (±0.053)
BayesB	3.735 (±0.219)	0.044 (±0.053)
BayesC	3.732 (±0.231)	0.043 (±0.043)
LASSO	3.719 (±0.220)	0.046 (±0.055)
BRR	3.723 (±0.257)	0.055 (±0.054)

**Table 2 plants-14-01422-t002:** Ancestry and number of seeds obtained for 28 self-fertilized genotypes of *Eucalyptus* spp.

Genotype	Ancestry			#Total Seeds	#Selfed	#Crossed
	*E. urophylla*	*E. grandis*	Unknown			
GEN1	0.500	0.500	0.000	28	12	16
GEN2	0.500	0.500	0.000	4	3	1
GEN3	0.500	0.492	0.008	19	19	0
GEN4	0.476	0.448	0.076	33	9	24
GEN5	0.487	0.498	0.015	33	29	4
GEN6	0.080	0.912	0.007	1	1	0
GEN7	0.495	0.500	0.005	33	13	20
GEN8	0.499	0.496	0.005	19	10	9
GEN9	0.380	0.609	0.011	3	2	1
GEN10	0.496	0.500	0.004	14	14	0
GEN11	0.500	0.495	0.004	31	28	3
GEN12	0.475	0.474	0.050	27	6	21
GEN13	0.455	0.455	0.090	33	1	32
GEN14	0.493	0.492	0.015	16	15	1
GEN15	0.510	0.475	0.015	3	2	1
GEN16	0.506	0.487	0.006	31	13	18
GEN17	0.498	0.496	0.006	1	1	0
GEN18	0.596	0.351	0.053	3	0	3
GEN19	0.600	0.303	0.096	30	27	3
GEN20	0.247	0.246	0.507	29	8	21
GEN21	0.494	0.500	0.006	3	3	0
GEN22	0.478	0.503	0.019	20	8	12
GEN23	0.498	0.496	0.006	17	17	0
GEN24	0.047	0.443	0.511	1	1	0
GEN25	0.502	0.496	0.001	32	32	0
GEN26	0.405	0.583	0.012	29	26	3
GEN27	0.463	0.535	0.002	17	15	2
GEN28	0.504	0.496	0.000	13	11	2

*E.*: *Eucalyptus*; GEN: Genotype.

**Table 3 plants-14-01422-t003:** Difference in priors between frequentist and Bayesian models used for phenotype prediction in a self-fertilized population of *Eucalyptus* spp.

Type	Model	Prior/Distribution
Frequentist	ABLUP	Additive Gaussian effects
	GBLUP	Additive Gaussian effects
	GBLUP-AD	Additive and dominance Gaussian effects
	HBLUP	Combination of priors from A and G
Bayesian	BRR	Gaussian prior for all markers
	BayesA	t-distribution (or scaled-t) for marker effects
	BayesB	Mixture: probability p of zero effect and (1-p) t or normal
	BayesC	Mixture: probability p of zero effect and (1-p) normal
	Bayes Lasso	Laplace (L1) prior

p: the probability of a null effect of the marker; L1: penalty associated with Laplace prior.

## Data Availability

The authors do not have permission to share the data.

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
