# Peer review of "Genomic Prediction in a Self-Fertilized Progenies of Eucalyptus spp."

_plants, 2025, doi:10.3390/plants14101422_

Round 1
Reviewer 1 Report
Comments and Suggestions for Authors
Clear mistake: 2.1.2 ....Due to limited number of seeds....., only 20 out of the 28 parents were used, with 30 seeds per parent selected, total 600 individuals.
Please see back at Table 1, only 9 Genotype have 30 or more seeds, where do you get 600 individuals???
You mentioned paternity test, but what is the result? I have not seen the data!
Comments on the Quality of English LanguageEnglish is OK
Author Response
All recommendations have been incorporated. Indeed, there was an error in the reported number of individuals in the experiments, which has now been corrected in the text to reflect the actual number of individuals, based on the available seed count. Regarding the paternity test, we have provided details on the type of marker used, as well as the number of available individuals (522) with molecular data.
Reviewer 2 Report
Comments and Suggestions for Authors
Report on "Genomic prediction in a self-fertilized progenies of Eucalyptus spp"
In this study, the authors conducted a trial of genomic prediction on eucalyptus, which includes important forestry tree species.
In eucalyptus, useful clones are produced by hybridization for commercial use.
From this perspective, there are few examples of trials of genomic prediction on closely related populations that have selfed, and it is expected that appropriate analysis can provide important insights.
However, the description of the method and the approach have the following concerns:
Line 100: This is the first time that a genealogy estimation based on markers is described in this paper. What kind of genetic markers were used? The legend of Fig. 1 provides supporting information, but an explanation is required in the text.
In the field test, were the seeds germinated and the resulting seedlings grown with clones of the parent? Or were the seedlings cloned by mini-cuttings or other methods and used in the growth test?
Were the seeds derived from self-pollination induced by applying a breeding bag to the same individual? Or was it artificial crossing? Or, even if it was open pollination, were they naturally self-pollinated and the seeds were identified later by genotyping?
The definition of MSE in Equation 18 seems strange. It's not squared, is it? The difference Σ can be negative, so an imaginary number is generated in the current equation. Please check the rigor of each equation again.
Also, the g used in the genomic prediction model above is different from the g in Equations 17 and 18 here, right? Using the same symbol for different definitions in one paper is confusing. Perhaps y, which is used as a response variable, is more appropriate.
After reading the result L269, I have a question. Was this self-pollination artificially targeted? If so, the 37% cross-pollination seems like a lot of contamination.
Or did parent-child analysis of seeds obtained in a large scale reveal that nearly 63% of the seeds were self-pollinated?
The method does not fully describe the aims and test process of these pollination systems.
Line 285 And in other places, please use terms such as prediction ability, prediction accuracy, and prediction capacity strictly and with their definitions.
The most common method is to use the correlation coefficient between the predicted value and the observed value in cross-validation as the prediction accuracy.
What does NA mean in the legend of Figure 5? Please review.
Table 3. What is EQM? It does not match the description in the table or the method text.
I also read the results of the genomic selection section from L284, but I still don't understand. In the method section, it says that the parent generation commercial clones were also included in the cultivation test, but were these included in the training and validation datasets for the genomic prediction model? If cuttings and seedlings are planted together, the difference between cuttings and seedlings should be included as a fixed effect. This is because the difference in the propagation method of cuttings or seedlings can affect the average performance (e.g. Matsushita 2025 in For Ecol Manage).
I also suspect that the predictive ability of the model would be improved if the parent generation was included in the training set. In particular, if the model treats the marker effect as a direct random effect, such as RIdge or Lasso, it would be possible to compare each SNP position between the parent and the offspring and estimate its direct effect.
In the case of selfing, if a marker position that was heterozygous in the parent generation becomes homozygous in the progeny, the effect can be estimated more directly.
Discussion and conclusion,
it seems that in recent forest tree breeding programs, including eucalyptus, higher accuracy has been achieved in genomic prediction, right?
The discussion does not mention much about the level of progress in genomic prediction in those previous recent studies.
This may be because the results of this study only had a relatively low predictive ability of around 0.48, in general.
As mentioned in the introduction, it was expected that a higher prediction rate would be obtained given the number of individuals by trying training and validation in a more closely related population, but it would be good to consider in more depth the results that this was not the case.
Author Response
Q1: Report on "Genomic prediction in a self-fertilized progenies of Eucalyptus spp" In this study, the authors conducted a trial of genomic prediction on eucalyptus, which includes important forestry tree species.
In eucalyptus, useful clones are produced by hybridization for commercial use.
From this perspective, there are few examples of trials of genomic prediction on closely related populations that have selfed, and it is expected that appropriate analysis can provide important insights.
However, the description of the method and the approach have the following concerns:
Line 100: This is the first time that a genealogy estimation based on markers is described in this paper. What kind of genetic markers were used? The legend of Fig. 1 provides supporting information, but an explanation is required in the text.
R1: In the paternity analysis, information regarding the type of marker (SNP), its quantity, and the number of individuals (522) used in the analysis has been described in the text, as requested.
Q2: In the field test, were the seeds germinated and the resulting seedlings grown with clones of the parent? Or were the seedlings cloned by mini-cuttings or other methods and used in the growth test?
R2: The seedlings used in the field trial were initially grown in a nursery and later transplanted to the field. The control clones, on the other hand, were produced through mini-stumps and were planted alongside the self-pollinated seedlings.
Q3: Were the seeds derived from self-pollination induced by applying a breeding bag to the same individual? Or was it artificial crossing? Or, even if it was open pollination, were they naturally self-pollinated and the seeds were identified later by genotyping?
R3: For the self-pollination process, the selected parents, which exhibited desirable genetic traits, were cloned and planted in the hybridization orchard. This location provides ideal conditions for performing both crossbreeding and self-pollination in a controlled manner, minimizing the risk of external contamination. After the plants were established, they were subjected to the artificial self-pollination process, where flowers were isolated and pollinated with their own pollen, ensuring that self-pollination occurred effectively. This procedure was carefully monitored to ensure the success of internal fertilization, resulting in the formation of self-pollinated seeds, which were then collected for subsequent analysis and use in the experiment.
Q4: The definition of MSE in Equation 18 seems strange. It's not squared, is it? The difference Σ can be negative, so an imaginary number is generated in the current equation. Please check the rigor of each equation again. Also, the g used in the genomic prediction model above is different from the g in Equations 17 and 18 here, right? Using the same symbol for different definitions in one paper is confusing. Perhaps y, which is used as a response variable, is more appropriate.
R1: The formula used for the mean squared error did indeed contain an error, which has been properly corrected with the correct formula. It is important to note that the calculation was performed correctly, and the error only occurred when describing it in the text. Regarding the suggestion to change the notation from "g" to "y" to avoid potential confusion in the reading, we appreciate the recommendation and have taken it into consideration.
Q5: After reading the result L269, I have a question. Was this self-pollination artificially targeted? If so, the 37% cross-pollination seems like a lot of contamination.
Or did parent-child analysis of seeds obtained in a large-scale reveal that nearly 63% of the seeds were self-pollinated? The method does not fully describe the aims and test process of these pollination systems.
R1: Yes, the self-pollination was artificially induced in the selected parents. Therefore, this high level of contamination is primarily associated with a failure in the self-pollination process, allowing pollen from nearby plants in the hybridization orchard to cross-pollinate. Furthermore, since the paternity test was conducted only after the individuals were already planted in the field, it was not possible to determine whether the individual was self-pollinated or cross-pollinated before the experiment was implemented. One must also consider the resistance to self-pollination present in the reproductive system of the tested species, both at pre-zygotic and post-zygotic stages.
Q6: I also read the results of the genomic selection section from L284, but I still don't understand. In the method section, it says that the parent generation commercial clones were also included in the cultivation test, but were these included in the training and validation datasets for the genomic prediction model?
I also suspect that the predictive ability of the model would be improved if the parent generation was included in the training set. In particular, if the model treats the marker effect as a direct random effect, such as Ridge or Lasso, it would be possible to compare each SNP position between the parent and the offspring and estimate its direct effect.
In the case of selfing, if a marker position that was heterozygous in the parent generation becomes homozygous in the progeny, the effect can be estimated more directly.
R6: In the cross-validation analysis, the parents were included within the training population. In a previous analysis not presented in the article, this information was not used, resulting in much lower predictive capacities. Suspecting an effect between parents and their respective offspring, and in order to increase our small sample size, we decided to use parental information to estimate the predictive accuracy. Therefore, we believe that the higher predictive capacities were not achieved, primarily due to the limited sample size.
Q7: It seems that in recent forest tree breeding programs, including eucalyptus, higher accuracy has been achieved in genomic prediction, right? The discussion does not mention much about the level of progress in genomic prediction in those previous recent studies. This may be because the results of this study only had a relatively low predictive ability of around 0.48, in general. As mentioned in the introduction, it was expected that a higher prediction rate would be obtained given the number of individuals by trying training and validation in a more closely related population, but it would be good to consider in more depth the results that this was not the case
R7: All considerations for the discussion of the results have been incorporated, providing a broader perspective on previous research related to genomic selection applied to Eucalyptus improvement. The aim was to highlight the differences between our study and those conducted previously.
Reviewer 3 Report
Comments and Suggestions for Authors
The manuscript entitled "Genomic Prediction in a Self-Fertilized Progenies of Eucalyptus spp." presents a study on genomic selection in Eucalyptus focusing on self-fertilized populations. The research aims to improve breeding efficiency by applying genomic prediction models to estimate genomic breeding values (GEBVs). The study addresses a significant issue in forestry genetics, emphasizing genomic selection to accelerate breeding cycles in Eucalyptus. The manuscript employs various genomic prediction models, including frequentist (GBLUP, HBLUP, ABLUP) and Bayesian (BayesA, BayesB, etc.) approaches, providing a robust comparative analysis. The results are well-documented, with appropriate use of statistical metrics. The research highlights the advantages of genomic selection over traditional breeding methods and demonstrates its applicability in forest tree breeding.
Minor remarks:
Line 101 – “a paternity analysis was conducted using Cervus software”
It needs to indicate the software version and references.
Line 131 – “were genotyped using the Eucalyptus 60K chip”
Is it a commercial chip or developed by Silva-Junior ?
Discussing how the results align or differ from prior research would strengthen the manuscript.
Author Response
Reviwer 3: The manuscript entitled "Genomic Prediction in a Self-Fertilized Progenies of Eucalyptus spp." presents a study on genomic selection in Eucalyptus focusing on self-fertilized populations. The research aims to improve breeding efficiency by applying genomic prediction models to estimate genomic breeding values (GEBVs). The study addresses a significant issue in forestry genetics, emphasizing genomic selection to accelerate breeding cycles in Eucalyptus. The manuscript employs various genomic prediction models, including frequentist (GBLUP, HBLUP, ABLUP) and Bayesian (BayesA, BayesB, etc.) approaches, providing a robust comparative analysis. The results are well-documented, with appropriate use of statistical metrics. The research highlights the advantages of genomic selection over traditional breeding methods and demonstrates its applicability in forest tree breeding.
Minor remarks:
Line 101 – “a paternity analysis was conducted using Cervus software”
It needs to indicate the software version and references.
R: This has been corrected.
Line 131 – “were genotyped using the Eucalyptus 60K chip”
Is it a commercial chip or developed by Silva-Junior?
R: This has been informed.
Discussing how the results align or differ from prior research would strengthen the manuscript.
R: We greatly appreciate your comments, which were essential in improving our manuscript and led to a significant enhancement. Thank you very much. In response to your observations, all of them have been incorporated. The version of the software used has been presented, along with the reference to the developer of the software in question. The marker chip used was the Eucalyptus 60K chip, a commercial chip developed by Silva-Junior, and the necessary modification has been made to clarify this information. Finally, additions were made to the discussion section, providing a broader view of how our research differs from others on the topic, including results from those studies for comparative purposes.
Round 2
Reviewer 2 Report
Comments and Suggestions for Authors
Based on the reviewers' comments, the authors have revised the manuscript.
However, some equations have not been bolded and italicized properly (e.g., equation 17), so please check them again.
Author Response
Thank you for your valuable feedback. We have carefully reviewed the manuscript and corrected the formatting of the Equation 17, ensuring that all are now properly bolded and italicized as required.